# Textbook Outcomes for Retroperitoneal Sarcoma Resection: A Multi-Centre Review

**DOI:** 10.3390/curroncol32060364

**Published:** 2025-06-19

**Authors:** Skyle Murphy, Christopher Allan, Andrew Barbour, Victoria Donoghue, B. Mark Smithers

**Affiliations:** 1Princess Alexandra Hospital, Brisbane, QLD 4102, Australia; 2School of Medicine, University of Queensland, Brisbane, QLD 4072, Australia; 3Mater Hospital, Brisbane, QLD 4101, Australia; 4Cancer Alliance Queensland, Woolloongabba, QLD 4102, Australia

**Keywords:** textbook outcome, surgery, resection, prognostic factors, survival

## Abstract

For patients with retroperitoneal sarcomas (RPSs), en-bloc resection with macroscopically negative margins remains the only potentially curative treatment. Textbook outcomes (TOs) are composite measures developed to compare ideal surgical outcomes for complex oncologic resections. The aims of this study were as follows: to define TO for RPS resections; to investigate the impact of treating service and other variables on TO; and to investigate the impact of treating service on achieving a TO. Population-based data from the Queensland Oncology Repository (QOR) was used to perform a retrospective review of all adult patients who underwent resection for primary RPS in Queensland between 2012 and 2022. TO was defined as follows: en-bloc resection; macroscopically negative margins; no unplanned ICU admission, no Clavien–Dindo III or greater complications; hospital length of stay of 14 days or less; no readmission within 30 days; and no 90-day mortality. A TO was achieved in 82 (56.94%) of the 144 patients included in the study. A high-grade histological subtype, the resection of three or more contiguous organs, major vascular resection and treatment outside of a high-volume sarcoma centre (HVSC) were significant negative predictors of achieving TOs (*p* < 0.05). On multivariate analysis, treatment at a high-volume sarcoma centre was independently associated with a 2.6-fold increase in TO (1.18–5.88, *p* = 0.02). Achieving a TO was associated with higher five-year DFS (61.5% vs. 41.3%, *p* = 0.03) and OS (76% vs. 59.4%, *p* = 0.02). In our state, TOs provide a measure of the quality of RPS resection across multiple health services, with patients treated at high-volume sarcoma centres more likely to achieve a TO. TO rates are associated with improved five-year DFS and OS.

## 1. Introduction

Retroperitoneal sarcomas (RPSs) are rare mesenchymal malignancies for which surgical resection remains the only potentially curative treatment. Oncological resection aims to achieve a macroscopically negative margin with en-bloc resection of involved contiguous organs and tissue. The completeness of surgical resection is an independent predictor of local recurrence and overall survival [1,2,3]. Surgical resection is technically challenging due to large tumour size, poorly defined disease extent and the proximity of critical retroperitoneal structures. The surgical approach should be individualised, taking into consideration tumour biology, expected peri-operative morbidity and the long-term functional impact of organ sacrifice [4]. The Transatlantic Australasian Sarcoma Working Group (TARPSWG) recommends the management of RPS by a multidisciplinary team at specialist sarcoma centres [5]. Although oncologically complete surgery is critical for long-term survival in patients with RPS, a comprehensive, standardised tool for assessing surgical quality is lacking.

Textbook outcomes (TOs) are composite measures aimed at identifying gold-standard surgical outcomes for complex oncological resections that allow for comparison across surgical services. First described by Kolfschoten et al. (2013) [6] for colorectal cancer resections, TOs have now been developed for several upper gastrointestinal and hepatobiliary malignancies [7,8,9,10,11]. The individual parameters used to define TOs are heterogeneous and disease-specific; however, all include direct or surrogate markers of technical proficiency, intra-operative complications, peri-operative course and short-term morbidity and mortality. Length of stay (LOS) is commonly utilised in TO analysis to indicate deviation from the ideal post-operative course. Previously, authors have used either expert opinion or a cohort percentile-based approach resulting in LOS endpoints from less than seven to twenty-one days, limiting the comparability and generalisability of results [6,7,8,12].

The primary aims of this study were as follows: to develop a TO for patients undergoing surgical resection for primary RPS, including the identification of an appropriate LOS threshold for population-based RPS TO analysis; and to assess the impact of achieving a TO on disease-free and overall survival. The secondary aim was to investigate the impact of the treating service on the achievement of a TO.

## 2. Materials and Methods

### 2.1. Study Population and Data Acquisition

A retrospective multi-centre review was performed on all adult patients who underwent surgical resection of primary RPS in Queensland, Australia, between 2012 and 2022. Patients were identified from the Queensland Oncology Repository (QOR), which is a state-wide cancer registry which collects demographic, diagnostic, treatment and patient outcome data. The notification of a pathological diagnosis of malignancy, including sarcoma (biopsy or surgical specimen), is mandatory for all public and private hospitals and pathology providers under Queensland legislation.

Histological inclusion and exclusion criteria were adopted from the TARPSWG consensus statements [5]. Excluded from analysis were the following: primary visceral, peritoneal or abdominal wall sarcomas; metastatic RPS; recurrent RPS; and emergency surgery performed for tumour rupture or haemorrhage.

### 2.2. Textbook Outcome Definition

TO was defined as the achievement of seven clinical and pathological parameters: en-bloc surgical resection, complete macroscopic resection, no unplanned ICU admission, no Clavien–Dindo III or greater complications, no prolonged hospital length of stay, no readmission within 30 days and no 90-day mortality. Optimal surgical technique was defined as the en-bloc resection of a tumour with involved contiguous organs. Macroscopically complete resection was defined as R0/1 resection on the basis of intra-operative assessment and specimen histology.

### 2.3. Length-of-Stay Analysis

An LOS threshold of 14 days or less was selected based on a review of the existing literature and preliminary analysis of the study cohort. Previously described LOS thresholds of 50th percentile (7 days), 75th percentile (12 days), 14 days and 21 days were evaluated [6,8,13,14]. As less than 5% of the study cohort had an LOS of 13–14 days, final analysis was limited to the more clinically applicable weekly intervals (Figure A1). Preliminary analysis using box plots was performed to explore the relationship between demographic factors, the complexity of surgical resection, inpatient morbidity and LOS (Figure A2 and Figure A3). A statistical comparison of LOS thresholds was performed using the Chi-squared test. Based on these findings, an LOS threshold of 14 days was adopted for inclusion in the definition of a textbook outcome.

### 2.4. Surgical Service Definition

In the absence of standardised definition, treating service was defined by three variables: the presence of surgeons with sarcoma expertise; hospital service capability; and median annual RPS resection volume. Surgeons considered to have sarcoma expertise were those with post-graduate training and clinical experience in soft-tissue oncology, registration with relevant clinical associations and regular participation in sarcoma multidisciplinary team (MDT) meetings. Hospital service capability was defined by the Australian Institute of Health and Welfare Hospital Peer Group (2015) [15]. High-service-capability hospitals include tertiary referral hospitals, public group A hospitals and private group A hospitals. These hospital categories provide a broad range of services including the presence of a 24 h emergency department, intensive care unit, interventional radiology services and high-level oncology and radiation facilities. A high-volume service was defined by a median annual resection volume of five cases or more [16,17]. Based on this, hospitals were categorised as high-volume sarcoma centres (HVSCs) or non-HVSCs. To further assess the impact of median annual resection volume and other service characteristics, non-HVSC services were stratified into two groups. Low-volume sarcoma centres (LVSCs) were defined by the presence of sarcoma specialists, a high hospital service capability and a median annual resection volume of fewer than five cases. Other centres were characterised by the absence of specialist sarcoma services.

### 2.5. Statistical Analysis

The association between TO and demographic, histological, surgical and treating service variables was analysed using Chi-square and Wilcoxon–Mann–Whitney tests. Univariate logistic regression was used to identify covariates associated with achieving a TO. Multivariate logistic regression was used to assess the independent contribution of gender, tumour histology, tumour size, multi-visceral resection, major vascular resection and surgical service on TO. Differences in disease-free and overall survival were compared using Kaplan–Meier curves and the log-rank test. A *p* value of <0.05 was considered statistically significant. Statistical tests were performed using R Statistical Software (version 4.4.2, R Core Team, Vienna, Austria, 2024).

## 3. Results

### 3.1. Study Cohort and Treating Service Characteristics

There were 144 patients who underwent resection for primary RPS during the study period. The median age was 62 years (range 26–83), and 79 (54.9%) patients were male (Table 1). The most prevalent tumour histologies were as follows: dedifferentiated liposarcoma (DDLPS) (42%); well-differentiated liposarcoma (WDLPS) (26%); and leiomyosarcoma (LMS) (22%). The median tumour size was 146 mm (range 27–460), and 4% of tumours displayed multifocality. Less than a quarter (30) of tumours originated in the pelvis.

Patients were treated across 20 facilities, of which 17 (85%) were concentrated in South East Queensland, 14 (70%) were private hospitals, and 14 (70%) were principal referrals hospitals or acute group A hospitals. Specialist sarcoma services were available at six (30%) centres.

In total, 77 (53%) patients underwent surgical resection at a single HVSC, which had a median annual resection rate of six cases per annum. Of the patients who underwent resection at other institutions, 22 (33%) were treated at LVSCs and 45 (31.2%) at other centres.

### 3.2. LOS Analysis

Preliminary LOS analysis revealed no significant differences in patient age, gender, tumour histology or tumour size between LOS thresholds of 7, 14 and 21 days (Table A1). An LOS greater than 14 days was associated with a statistically significant increase in surgical complexity, the incidence of minor and major medical and surgical complications and unplanned ICU admissions (*p* < 0.05). Based on these findings, an LOS threshold of 14 days was adopted for inclusion in the definition of a textbook outcome.

### 3.3. Factors Associated with TO Achievement

A TO was achieved in a total of 82 (56.94%) cases. There were no statistically significant differences in TOs based on age, gender, tumour size, anatomical location, multifocality or the receipt of pre-operative biopsy, multidisciplinary team (MDT) discussion, neoadjuvant radiotherapy or adjuvant radiotherapy (*p* > 0.05) (Table 1). Tumour histology was significantly associated with TOs, with 29 (76.3%) patients with WDLPS attaining a TO compared to 30 (55%) patients with DDLPS and 11 (35.5%) patients with LMS (*p* = 0.002). Major multi-visceral resection (three or more organs) (31.6%, *p* 0.042) and major vascular resection (36.4%, *p* = 0.004) were negatively associated with achieving a TO.

Surgical service was also associated with higher rates of TOs, with 50 (61.0%) patients treated at HVSCs having a TO as compared to 32 (39.0%) treated elsewhere (*p* = 0.04). Of those treated at non-HVSCs, 13 (59.1%) patients treated at LVSCs and 19 (42.2%) patients treated at other centres achieved a TO (Figure 1). Patients treated at HVSCs were more likely to have a pre-operative core biopsy (75% vs. 28%, *p* < 0.001), be the subject of a pre-operative MDT presentation (74% vs. 18%, *p* < 0.001) and be offered neoadjuvant radiotherapy (13% vs. 0%, *p* < 0.002).

The individual frequency and cumulative frequency of TO parameters are depicted in Figure 2a. The most frequent reasons for failure to meet TO criteria were prolonged LOS, affecting 25 (17.4%) patients, and 30-day readmission, affecting 24 (16.7%) patients. Peri-operative morbidity was low, with eleven (7.6%) having a Clavien–Dindo grade III or greater complication and eight (5.6%) requiring ICU readmission. Ninety-day mortality was less than 1%.

A comparison of TO parameter attainment between HVSCs and other hospitals demonstrated significant differences in surgical approach and the completeness of resection (Figure 2b). Limited surgical resection, including four piecemeal resections or intra-operative tumour rupture, occurred in 15 (22.3%) patients treated outside of an HVSC. Consequently, incomplete macroscopic resection occurred in nine (13.4%) non-HVSC cases. In contrast, only one patient at an HVSC had an R2 resection after failing to proceed to second-stage resection for multifocal disease.

### 3.4. Univariate and Multivariate Analysis

On univariate logistic regression the likelihood of achieving a TO was lower in patients with DDLPS, LMS and UPS compared to WDLPS (Table 2). Tumour size greater than 20 cm was associated with a 4.3-fold increase in the odds of achieving a TO, although not statistically significantly. Of the 45 patients with tumours greater than 20cm, a third (33.3%) had WDLPS and 55% were treated at an HVSC. Patients requiring multi-visceral resection of three or more organs showed a trend towards reduced rates of TOs (*p* = 0.083). Patients who required major vascular resection had a 5.3-fold reduction in TOs. Conversely, patients treated at an HVSC had a twofold increase in their odds of achieving a TO.

On multivariate analysis, surgical service remained an independent predictor of TO after adjusting for potential confounders including gender, tumour histology, multi-visceral resection and major vascular resection.

### 3.5. Survival Analysis

The median follow-up period was 4 years and 8 months. Disease recurrence occurred in 69 (48%) patients at a median of 19 months. Local failure occurred in 38 (55.1%) cases. Overall five-year DFS was 53% and was significantly higher for patients who achieved a TO (61.5% vs. 41.3%, *p* = 0.029) (Figure 3a). Five-year OS was 68.5% and was higher in patients who achieved a TO (59.35% vs. 76.04%, *p =* 0.022) (Figure 3b).

## 4. Discussion

This study establishes a model for assessing TOs in patients undergoing surgical resection for primary RPS. In a population-based cohort we assessed 144 patients using seven parameters related to the completion of resection and the impact of complications from what can be a complex major surgery. The overall TO rate in this series was 56.94%. An LOS greater than 14 days and readmission within 30 days from discharge were the two most common reasons for patients failing to achieve a TO. On multivariate analysis, a dedifferentiated liposarcoma histological subtype and increasing resection complexity were associated with a lower likelihood of achieving a TO. Despite potential referral bias towards more complex cases, treatment in an HVSC was an independent predictor for achieving a TO. The achievement of a TO was associated with superior five-year DFS and OS.

The use of surgical quality assessment tools beyond single-outcome measures such as peri-operative complications or mortality has gained momentum in recent years [18]. Reported TO rates for complex oncological gastrointestinal resections vary considerably, including the following: esophagectomy (29.7–50.8%) [7,19,20], gastrectomy (22.7–51%) [7,21,22], pancreaticoduodenectomy (16.8–60.3%) [9,11,12] and hepatectomy (15.8–77%%) [11,23,24,25].

The individual metrics and number of criteria included in TO definitions demonstrate substantial heterogeneity across published studies [8,12]. Nonetheless, most definitions include measures of oncological resection, post-operative morbidity, LOS and operative mortality.

In the present study, surgical technique, defined as the en-bloc resection of a tumour with involved contiguous organs, was included based on the following: the emphasis on radical resection in landmark TO studies [6,7] and institutional and individual variability in the extent of resection performed for primary RPS. Both microscopically negative (R0) and involved (R1) margins were considered complete due to the technical challenges of resection, impractical pathological assessment of all microscopic tumour margins and comparable overall survival [5]. Unplanned ICU admission and Clavien–Dindo III or greater complications were included, as these outcomes are reliably recorded at an institutional level and therefore allow for reliable replication in both retrospective and prospective multi-centre analyses. Readmission within 30 days and mortality within 90 days were selected endpoints due to their routine use in clinical practice and high concordance with the TO literature [8,12].

LOS is a key component of TO analysis. LOS inclusion acts as a surrogate marker for deviations from expected post-operative trajectory not otherwise captured by other measures of post-operative complications. LOS also provides a metric for hospital resource utilisation and financial expenditure, providing an important measure for institutional cost-effectiveness. LOS is a multifactorial outcome influenced by patient characteristics, disease and surgical complexity, cultural factors and institutional practices. Despite its regular inclusion, no standardised methodology for defining an optimal LOS has been established in the TO literature. Commonly employed methodologies include expert opinion and cohort percentile-based approaches (commonly 50th or 75th percentile). As a result, reported LOS thresholds range from 7 to 21 days [6,7,8,12]. The selection of a lower-range LOS may fail to appropriately reflect the clinical complexity managed by high-volume specialist centres and may disproportionately exclude patients with advanced disease from achieving a TO. In the current series, preliminary analysis identified that day 14 was the inflexion point at which deviation from the expected post-operative course occurred. As such, it serves as a clinically meaningful and evidence-based threshold for defining a prolonged LOS.

To date, only two studies have investigated TOs in patients undergoing resection for RPS. Moris et al. (2020) recently performed a population-based retrospective analysis using the following RPS TO definition: length of stay <75th percentile, no readmission within 30 days, survival > 90 days and gross negative margins. A TO was achieved in 54% of the 11,032 primary RPS resections [13]. This TO definition did not include a measure of surgical technique, and notably, only 19.3% of patients underwent radical resection, 41.2% underwent limited resection and 39.5% underwent local excision. Consequently, failure to achieve macroscopically complete resection occurred in 55.9% of patients, compared with an R0/1 resection rate of 93.06% in our series. A more stringent definition was proposed by Wiseman et al., (2020) who defined a TO as the absence of the following: R2 resection, Clavien–Dindo ≥ II complication, transfusion of packed red blood cells peri- or post-operatively, reoperation, hospital LOS > 50th percentile, readmission within 90 days, non-home discharge and mortality within 90 days [14]. When these criteria were applied to 627 patients treated at high-volume academic centres, 34.9% achieved a TO. The rate of peri-operative complications was higher than those reported in both the current series (7.6%) and the literature (16.4% vs. 33.2%) [26]. This discrepancy is likely attributable to variations in definitional criteria of complications (Clavien–Dindo II vs. III complication), as tumour size over 20cm (23.0%) and high-grade tumour histology (64.4%) were equivalent to in our cohort. Notably, 90-day mortality was also lower in our current series at 1.2%, compared to 3.2%.

Clinical guidelines recommend the management of RPS by multidisciplinary teams at specialised sarcoma centres [5]. Benefits of treatment at specialist centres include the following: dedicated pre-operative radiological and histopathology examination; the availability of subspecialty expertise including upper gastrointestinal, hepatobiliary and vascular surgery; and access to multimodal therapies including clinical trials. With regard to pre-operative assessment, the Transatlantic Australasian Retroperitoneal Sarcoma Working Group (TARPSWG) recommends cross-sectional imaging, core needle biopsy, and multidisciplinary team (MDT) input as essential components of best-practice care. More recently, the group has also advocated for the consideration of neoadjuvant radiotherapy in histological subtypes with a high risk of local recurrence, particularly well-differentiated liposarcoma (WDLPS) and low-grade dedifferentiated liposarcoma (DDLPS). Deviations from these recommendations were observed in the present study. This is attributable, in part, to this study’s extended recruitment period, which began prior to the widespread implementation of the TARPSWG guidelines. In earlier cases, operative decisions were made on the basis of pathognomonic radiological findings without histological confirmation. Additionally, adherence to pre-operative assessment guidelines was significantly lower among patients treated at non-HVSCs, highlighting ongoing variability in the implementation of best-practice guidelines across institutions and supporting the role of the centralisation of care.

Contemporary multi-centre retrospective cohort studies have demonstrated that the treatment of primary RPS at high-volume and/or specialist sarcoma centres is associated with an increased rate of surgical resection, complete macroscopic resection (R0/1), an increased use of peri-operative radiotherapy and/or chemotherapy, decreased peri-operative mortality, decreased 2- and 5-year local recurrence and increased median and 5-year overall survival [27,28,29,30,31]. In the present series, treating service was independently associated with achieving a TO, with surgery performed at an HVSC conferring a 2.6-fold increase in the odds of achieving a TO. A clear definition of specialist sarcoma services is lacking. Specialist sarcoma centres have been variably defined in the literature using annual sarcoma-related hospital admissions [27], annual RPS resection volumes [28,29], referral service [30], academic affiliation [28] and clinical network membership [31]. Recently, Samà et al. (2024) identified a learning curve threshold of 46 cases for a single surgeon to achieve competency in RPS surgery, underscoring the technical complexity inherent in multi-visceral sarcoma resections [32]. In the present study, a multidimensional definition of surgical service was employed, including the presence of a sarcoma surgeon, hospital service capability and a median annual resection volume of greater than five. This number was selected as a volume threshold as it has been used in comparable Australian studies for other complex oncological resections such as gastrectomy [16,17].

The clinical utility of TO is highlighted by its strong association with long-term survival. In the present study, achieving a TO conferred a 49% relative increase in five-year DFS and 28.1% increase in OS (*p* < 0.05). In the only other study which has assessed TO for RPS and survival, TO was associated with improved median recurrence-free survival (8.5 years compared to 3.8 years, *p* < 0.1) and OS (12.8 years compared to 6.4 years, *p* > 0.1) for patients with primary RPS [14].

Limitations of this study include the small sample size, which reflects the low incidence of STS and the geographic dispersion of the Australian population. Due to the retrospective nature of the study, comprehensive and consistent documentation of comorbidities was not available for all patients and therefore could not be included in the analysis. Similarly, minor (Clavien–Dindo grade I–II) complications were not systematically recorded, which precluded the use of cumulative morbidity scores such as the Comprehensive Complication Index (CCI). The CCI is a validated tool that has been shown to correlate strongly with key outcomes, including post-operative length of stay and total hospitalisation costs, in patients with retroperitoneal sarcoma [33]. In addition, the long recruitment period spanned a time during which the understanding of tumour biology evolved, surgical resection techniques became refined and the use of radiotherapy and chemotherapy changed due to emerging evidence from clinical trials. This study was also limited by a lack of standardised reporting of intra-operative resection status. Finally, long-term follow-up is required to identify patterns of disease recurrence, particularly for low-grade subtypes.

## 5. Conclusions

TO provides a composite metric that allows for the objective comparison of surgical quality across health services. In this study, treatment at HVSCs was independently associated with higher rates of TO achievement. Moreover, TO conferred a significant increase in five-year DFS and OS. This highlights the importance of multidisciplinary specialist care for patients with RPS.

## Figures and Tables

**Figure 1 curroncol-32-00364-f001:**
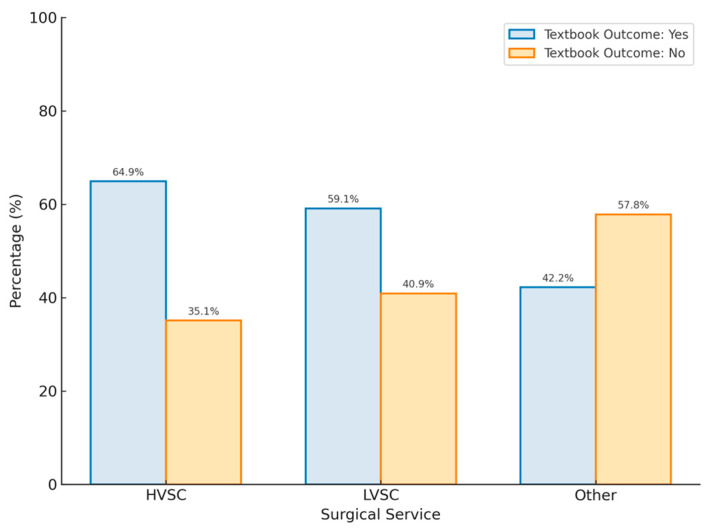
Proportion of patients who achieved TO stratified by surgical service, which was defined as high-volume sarcoma centres (HVSCs), low-volume sarcoma centres (LVSCs) and other.

**Figure 2 curroncol-32-00364-f002:**
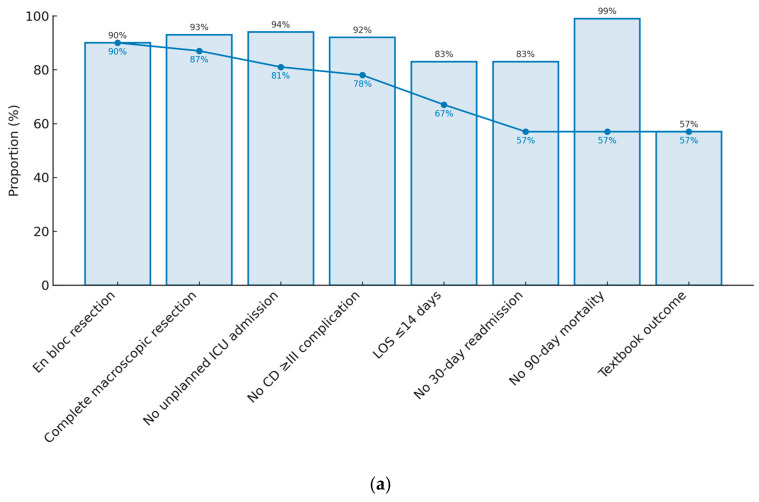
Individual and chronological cumulative distribution of TO parameters in (**a**) all patients undergoing surgical resection of primary RPS and (**b**) stratified by surgical service defined as high-volume sarcoma centre (HVSC) versus non-high-volume sarcoma centre (non-HVSC).

**Figure 3 curroncol-32-00364-f003:**
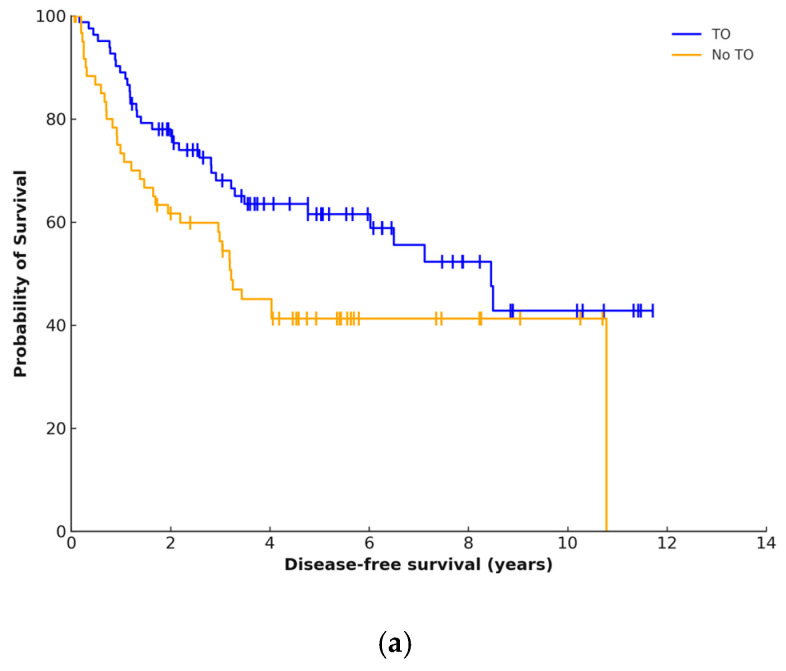
Kaplan–Meier plots for (**a**) disease-free survival and (**b**) overall survival stratified by textbook outcome (TO) achievement.

**Table 1 curroncol-32-00364-t001:** Demographic, tumour and treatment variables for patients with and without TO.

		Textbook Outcome		
	Total	Yes	No	*p* Value
Variable, *n* (%)	*n* = 144	*n* = 82	*n* = 62	
Age				0.35
<50	23 (16.0)	13 (15.9)	10 (16.1)	
50–70	83 (57.6)	51 (62.2)	32 (51.6)	
>70	38 (26.4)	18 (22.0)	20 (32.3)	
Gender				0.13
Male	79 (54.9)	40 (48.8)	39 (62.9)	
Female	65 (45.1)	42 (51.2)	23 (37.1)	
Histology				0.002
WDLPS	38 (26.4)	29 (35.4)	9 (14.5)	
DDLPS	60 (41.7)	33 (40.2)	27 (43.5)	
LMS	31 (21.5)	11 (13.4)	20 (32.3)	
UPS	7 (4.9)	2 (2.4)	5 (8.1)	
Other	8 (5.6)	7 (8.5)	1 (1.6)	
Tumour size				0.25
<5 cm	59 (41.0)	31 (37.8)	28 (45.2)	
5–10 cm	28 (19.4)	14 (17.1)	14 (22.6)	
10–20 cm	8 (5.6)	3 (3.7)	5 (8.1)	
>20 cm	45 (31.2)	31 (37.8)	14 (22.6)	
Unknown	4 (2.8)	3 (3.7)	1 (1.6)	
Anatomical location				0.33
Right retroperitoneal	47 (32.6)	28 (34.1)	17 (27.4)	
Left retroperitoneal	55 (38.2)	33 (40.2)	21 (33.9)	
Central	11 (0.6)	4 (0.4)	4 (0.4)	
Bilateral	1 (0.1)	0 (0)	0 (0)	
Pelvic	30 (20.8)	17 (20.7)	16 (25.8)	
Multifocality				
Yes	6 (4.2)	3 (3.7)	3 (4.8)	>0.99
No	138 (95.8)	79 (96.3)	59 (95.2)	
Pre-operative biopsy				0.61
Yes	77 (53.5)	42 (51.2)	35 (56.5)	
No	67 (46.5)	40 (48.8)	27 (43.5)	
Pre-operative MDT				0.13
Yes	69 (47.9)	44 (53.7)	25 (40.3)	
No	75 (52.1)	38 (46.3)	37 (59.7)	
Neoadjuvant radiotherapy				>0.99
Yes	10 (6.9)	6 (7.3)	4 (6.5)	
No	134 (93.1)	76 (92.7)	58 (93.5)	
Surgical service				0.04
HVSC	77 (53.5)	50 (61.0)	27 (43.5)	
Non-HVSC	67 (46.5)	32 (39.0)	35 (56.5)	<0.05
LVSC	22 (15.3)	13 (15.9)	9 (14.5)	
Other	45 (31.2)	19 (23.2)	26 (41.9)	
Number of organs resected				0.042
0	42 (29.2)	24 (29.1)	19 (30.7)	
1–2	83 (57.6)	52 (63.4)	30 (48.4)	
3+	19 (13.2)	6 (7.3)	13 (21)	
Major vascular resection				0.004
Yes	11 (11.8)	4 (4.9)	13 (21)	
No	127 (88.2)	78 (95.1)	49 (79)	
Adjuvant radiotherapy				
Yes	11 (7.6)	6 (7.3)	5 (8.1)	>0.99
No	133 (92.4)	76 (92.7)	57 (91.9)	

**Table 2 curroncol-32-00364-t002:** Univariate and multivariate logistic analysis to identify demographic, tumour, management and surgical factors predictive of TO. Multivariate analysis performed for all variables with a *p* value of <0.1 identified on univariate analysis.

Characteristic	Univariate			Multivariate		
	OR	(95% CI)	*p* Value	OR	(95% CI)	*p* Value
Age						
<50	-			-		
50–70	1.23	(0.48–3.12)	0.67			
>70	0.69	(0.24–1.96)	0.49			
Gender						
Male	-			-		
Female	1.78	(0.91–3.49)	0.09	2.03	(0.89–4.66)	0.09
Tumour histology						
WDLPS	-			-		
DDLPS	0.38	(0.15–0.94)	0.04	0.37	(0.13–1.07)	0.06
LMS	0.17	(0.06–0.49)	0.001	0.32	(0.08–1.28)	0.11
UPS	0.12	(0.02–0.75)	0.02	0.19	(0.02–1.49)	0.11
Other	2.17	(0.23–20.10)	0.49	2.65	(0.24–28.65)	0.42
Tumour size (cm)						
<5	-			-		
5–10	2.31	(0.48–11.12)	0.02	2.25	(0.34–14.78)	0.40
10–20	2.21	(0.51–9.70)	0.29	1.71	(0.29–9.94)	0.55
>20	4.29	(0.93–19.68)	0.06	2.96	(0.47–18.47)	0.25
Pre-operative biopsy						
No	-			-		
Yes	0.81	(0.42–1.57)	0.53			
Pre-operative MDT						
No	-			-		
Yes	1.71	(0.88–3.34)	0.11			
Pre-operative radiotherapy						
No	-			-		
Yes	1.14	(0.31–4.24)	0.84			
Multi-visceral resection						
0	-			-		
1–2	1.37	(0.65–2.91)	0.41	1.32	(0.50–3.49)	0.57
3+	0.37	(0.12–1.14)	0.08	0.29	(0.07–1.20)	0.09
Major vascular resection						
No	-			-		
Yes	0.19	(0.06–0.63)	0.006	0.19	(0.04–0.93)	0.04
Surgical service						
Non-HVSC	-			-		
HVSC	2.03	(1.04–3.96)	0.04	2.64	(1.18–5.88)	0.02

## Data Availability

Restrictions apply to the availability of these data. Data were obtained from Cancer Alliance Queensland. Requests to access the datasets should be directed to Cancer Alliance Queensland.

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
