# Peer review of "Textbook Outcomes for Retroperitoneal Sarcoma Resection: A Multi-Centre Review"

_curroncol, 2025, doi:10.3390/curroncol32060364_

Round 1
Reviewer 1 Report
Comments and Suggestions for Authors
In this study the authors aimed to define Textbook outcomes (TO) for retroperitoneal sarcoma (RPS) resections; to investigate the impact of treating service and other variables on TO; and to investigate the impact of treating service on achieving a TO.
TO are defined as composite measures aimed at identifying gold-standard surgical outcomes for complex oncological resections that allow for comparison across surgical services.
A retrospective review was performed using population-based data from the Queensland Oncology Repository (QOR) to examine all adult patients who underwent resection for RPS in Queensland between 2012 and 2022.
TO was finaly defined as: en-bloc resection; macroscopically negative margins; no unplanned ICU admission, no Clavien-Dindo III or greater complications; hospital length of stay of 14 days or less, no readmission within 30-days and no 90-day mortality.
TO was achieved in 82 (56.94%) of the 144 patients included in the study. High-grade histological subtype, resection of three or more contiguous organs, major vascular resection and treatment outside of a high-volume sarcoma centre (HVSC) were significant negative predictors of achieving TO (p<0.05).
Comments
1.Using surgical quality assessment tools that go beyond single outcome measures, such as peri-operative complications or mortality, is very important for patients and for the economic aspects of healthcare. This is because unqualified operations lead to complications, prolonged hospital stays and increased costs.
Multivariate analysis revealed that treatment at a high-volume sarcoma centre was independently associated with a 2.6-fold increase in TO (1.18–5.88, p = 0.02). This is a very important finding which supports the general trend towards centralised care for sarcoma patients at high-volume sarcoma centres.
However, other measures included in the TO, such as en bloc resection and LOS, depend not only on technical proficiency, but also on tumour histological type (i.e. different growth characteristics), tumour extension, and the general conditions and comorbidities of patients.
The authors did not find any correlation between TO and age; however, co-morbidities are not necessarily more prevalent in older patients. Many younger adults with obesity, diabetes or cardiovascular disease may require prolonged hospital stays, despite the centre's technical proficiency.
The TO defined here does not take into account the patients' other health conditions and may therefore be misinterpreted.
- Similarly, resection of three or more contiguous organs and major vascular resection were found to be negative risk factors. However, this is not a sign of poor service quality, but a necessity associated with the tumour's extension and localisation.
- Line 78: Macroscopically complete resection was defined as an R0/1 resection based on an intraoperative assessment and histological examination.
While pathological assessment of all microscopic tumour margins is challenging in RPS, patients with R1 resection more frequently receive post-operative radiotherapy (RT) and therefore have comparable overall survival rates to those with R0 resections. Post-operative therapy has not been analysed or taken into account in the calculation of disease-free survival (DFS) and overall survival (OS).
The authors should comment on this.
- Table 2
Only 53% of patients underwent a preoperative biopsy. A biopsy (also CT-guided tru-cut) is highly recommended for all tumours diagnosed on imaging only, and is a base for decision taken during MTD. MTD was not relevant to TO in this study. It is astonishing that two measures — biopsy and MTD — which are recommended as the standard of care for patients with tumours in general, are not relevant in this analysis.
- Line 147-149 The authors stated that tumour histology was associated with TO, rather than biopsy. However, this comparison is incorrect and biased since biopsies were only performed on 53% of patients and a final histological diagnosis was made for all tumours. Knowledge of histology facilitates resection planning; therefore, if all patients had undergone biopsy, the operation would have been planned more specifically according to histological type.
Authors should explain why biopsies were not performed and how the OP was planned without knowledge of the tumour type.
- Similarly, preoperative radiotherapy (RT) has recently been recommended as a measure to improve outcomes and facilitate surgery, but it was not found to correlate with outcomes in this analysis. The proportion of patients who received preoperative radiotherapy was very low (7%), so it is difficult to compare the results depending on this measure. This is especially the case given that the significance of preoperative radiotherapy depends on histology (i.e. it is associated with a better outcome in well-differentiated liposarcoma and G1-2 dedifferentiated liposarcoma), and the histology was not known in 47% of patients preoperatively.
- Line 238 Univariate and multivariate analyses revealed that high-grade histological subtypes were associated with a lower likelihood of achieving TO.
The Methods do not define high-grade histological subtypes.
- Line 279 “At the population level, these findings support the use of a 14-day LOS threshold in TO analyses, as LOS exceeding 14 days was associated with a significant increase in perioperative morbidity”.
It is obvious that LOS is prolonged by complications; otherwise, patients would be discharged. This statement can therefore be omitted.
Minor
51 utalised Typo: probably utilised.
Reviewer 2 Report
Comments and Suggestions for Authors
The manuscript is clearly written, with well-defined objectives and a solid methodological approach. Results are robustly analyzed and effectively presented, offering valuable insight into the prognostic significance of textbook outcomes in retroperitoneal sarcoma surgery. Minor revisions are required, including the addition of key references and correction of typographical errors for consistency and professionalism (e.g., “utalised” should be “utilised”)
Author Response
Comment 1: Minor revisions are required, including the addition of key references and correction of typographical errors for consistency and professionalism (e.g., “utalised” should be “utilised”)
Response 1:
We have made the assessed the manuscript for typographical errors and made the following changes – utilised and multidisciplinary.
The authors acknowledge the value of a standardized, comprehensive scoring system for postoperative complications, such as the Comprehensive Complication Index (CCI). This tool is particularly relevant in the context of retroperitoneal sarcoma, where the overall rate of major morbidity approaches 16%, and patients often experience multiple, heterogeneous complications. The superiority of CCI over the Clavien-Dindo (CD) classification in capturing the cumulative burden of morbidity has been demonstrated by Ruspi et al. (Eur J Surg Oncol, 2022), who showed stronger associations between CCI scores and important outcomes such as postoperative length of stay and total hospitalization costs. However, given the retrospective, population-based design of the current study, minor complications (CD grade I–II) were not systematically recorded, precluding the reliable calculation of CCI in this context. We agree that standardization of complication reporting, including adoption of tools like the CCI, is an important goal for future prospective studies evaluating outcomes in sarcoma surgery. We have acknowledge this limitation in the discussion section line 351-355.
We thank the reviewer for this insightful comment and have incorporated the suggested citation and perspective into the discussion. In the revised manuscript, we acknowledge the ongoing variability in how specialist sarcoma centres are defined and highlight the emerging role of surgeon-specific metrics discussion lines 333-336. This addition supports the argument that surgical specialization may be better defined not only by institutional volume but also by cumulative procedural experience, as demonstrated through learning curve analysis. This aligns with our study’s multidimensional definition of surgical service, which includes surgeon expertise, hospital capability, and resection volume.
Reviewer 3 Report
Comments and Suggestions for Authors
The authors should be congratulated on an excellent paper.
Just one line does not make sense (236-237)
LOS less than 14 days and no readmission after 30-days were the two most common reasons
for the lack of achieving a TO
Author Response
Comment 1: Just one line does not make sense (236-237) LOS less than 14 days and no readmission after 30-days were the two most common reasons for the lack of achieving a TO
Response 1: We thank the reviewer for the assessment. We have made the following correction - LOS greater than 14 days and re-admission within 30-days from discharge were the two most common reasons for patients failing to achieve a TO. Discussion lines 242-243.